# OpenReview forum: "SPLIT-VLM: Salience-Guided Partitioning towards Local Coverage for Importance-Aware Token Dropping in Vision-Language Models"
_ICLR.cc/2026/Conference — Submitted to ICLR 2026_

### Official Review · Reviewer_96wB · 2025-10-27

**Soundness:** 3
**Presentation:** 3
**Contribution:** 2
**Rating:** 6
**Confidence:** 4

**Summary:**

SPLIT is a theoretically grounded framework for efficient token reduction in large-scale vision–language models, jointly preserving salience and diversity while eliminating redundancy. By estimating token importance via temporal shifts, assigning adaptive region-level budgets, and selecting distinctive tokens, SPLIT achieves robust performance under severe token constraints. Experiments show that it outperforms prior methods on image and video benchmarks, maintaining high accuracy even with drastically reduced vision tokens.

**Strengths:**

1. The paper is clear and practical, proposing SPLIT to efficiently balance token importance in vision–language models. The manuscript is well-structured and presents the method and results in a coherent and accessible manner.

2. I personally find it very interesting to apply shift bias from temporal domain to token pruning.

3. Experimental validation is sufficient. The authors conduct comprehensive experiments on various tasks and show **improvements**, to validate the effectiveness of the method. Moreover, the ablation study is detailed, particularly in the efficiency analysis section.

**Weaknesses:**

1. In table 3, compared with GreedyPrune, SPLIT performs notably better under the 64-token setting. However, its advantage diminishes under the 128- and 192-token settings.

2. Equations 4 and 7 are missing periods.

3. Does the diversity-based selection in SPLIT overlap with previous methods, such as DivPrune and CDPruner[1]?

[1] Beyond Attention or Similarity: Maximizing Conditional Diversity for Token Pruning in MLLMs, NeurIPS 2025.

**Questions:**

See above Weaknesses.

---

> ### Author Response · Authors · 2025-11-20
> **Response to Reviewer 96wB**
>
> We thank Reviewer **96wB** for the helpful feedback. Please find our point-by-point response below.
>
> ### **A1) Limited performance improvement**
> >- Although SPLIT shows slightly lower performance only in a very small number of benchmarks and only under specific budget conditions, it maintains consistently stable performance across virtually all other settings and achieves substantially higher performance under small token budgets. In addition, SPLIT achieves the fastest latency in all evaluated conditions, and this advantage becomes more pronounced as the token budget increases. SPLIT shows a consistent strength: **better performance under smaller budgets and better latency under larger budgets**.
> > - Table F presents the latency across different token budgets.
>
> > **Table F** *LLaVA-Next-7B latency (88.9%)*
> | Method |Tokens| Total Time|Δ Time vs 320 |
> |-----------|-----------|----------------|---------------------|
> | GreedyPrune | 320 | 31:17 | - |
> | GreedyPrune | 640 | 32:08 | +0:51 |
> | GreedyPrune | 960 | 33:06 | +1:49 |
> | DART | 320 | 27:28 | - |
> | DART | 640 | 28:13 | +0:45 |
> | DART | 960 | 28:57 | +1:29 |
> | SPLIT | 320 | 25:52 | - |
> | SPLIT | 640 | 26:24 | +0:32 |
> | SPLIT | 960 | 26:58 | +1:06 |
>
> ---
> ### **A2) Equations- missing periods**
> >- Thank you for the careful checking. We have updated the revised manuscript accordingly.
>
> ---
> ### **A3) Overlap between SPLIT’s diversity selection and prior methods**
> >- DivPrune is the closest prior approach, but our selection differs. DivPrune uses a max–min pairwise distance objective, whereas SPLIT relies on mean and variance of the similarity distribution.
> >- CDPruner captures diversity well but relies on DPP, which introduces heavy computational overhead. Also, it requires an additional CLIP text encoder for instruction-conditioned similarity, **making it impractical for models like Qwen**.
> >- Table G1–5 reports image understanding performance and latency for various related methods, including CDPruner. We carefully examined the official GitHub repositories of all three methods and ensured their algorithms were faithfully reproduced. All comparisons were conducted under the same experimental environment, and we evaluated every method using the LMMs-eval.
>
> > **Table G-1** *LLaVA-1.5-7B — Full Token, 576 Tokens (100%)*
> | Method  | GQA   | MMB   | MMB-CN | MME     | POPE | SQA   | VQA (txt) | VQA (v2) | VizWiz | OCRBench | Ratio  |
> |-----|---|----|----|----|----|---|---|---|----|---|----|
> | Vanilla | 61.2  | 63.1  | 54.8   | 1477.7  | 85.4 | 68.1  | 48.7      | 77.4     | 53.8   | 205      | 100%   |
> >
> > **Table G-2** *LLaVA-1.5-7B — Retain 192 Tokens (66.7%)*
> | Method     | GQA  | MMB  | MMB-CN  | MME    | POPE | SQA  | VQA (txt) | VQA (v2) | VizWiz | OCRBench | Ratio        |
> |-----|----|------|-----|----|----|------|-----|-------|--------|----------|------|
> | Vispruner  | 58.6 | 61.5 | 53.6    | 1469.9 | 85.2 | 68.3 | 45.7      | 75.7     | 54.9   | 179      | 98.0%        |
> | Visionzip  | 58.1 | 61.7 | 54.3    | 1447.1 | 85.0 | 68.8 | 43.9      | 75.7     | 54.2   | 180      | 97.0%        |
> | CDPruner   | 59.8 | 62.7 | 55.6    | 1483.6 | 86.2 | 67.5 | 48.9      | 76.8     | 55.7   | 189      | 99.5%        |
> | SPLIT | 59.5 | 62.6 | 54.5    | 1479.5 | 85.6 | 68.1 | 46.9      | 76.1     | 55.3   | 192      | 99.3%        |
> >
> > **Table G-3** *LLaVA-1.5-7B — Retain 128 Tokens (77.8%)*
> | Method     | GQA | MMB | MMB-CN | MME    | POPE  | SQA  | VQA (txt) | VQA (v2) | VizWiz | OCRBench | Ratio        |
> |-----|----|-----|-----|------|------|------|---|-------|-----|---|---|
> | Vispruner  | 58.0| 61.0| 52.9    | 1445.7 | 85.7 | 67.8 | 43.6| 75.0     | 55.7   | 177| 96.7%|
> | Visionzip  | 56.9| 61.7| 53.6    | 1412.7 | 83.7 | 68.4 | 43.2| 74.5     | 55.5   | 184| 95.4%|
> | CDPruner   | 58.4 | 61.8 | 53.5  | 1458.0 | 85.6 | 66.8 | 45.2 | 75.1 | 55.2 | 180 | 97.4% |
> | SPLIT| 58.8| 62.5| 51.9    | 1455.2 | 85.5 | 68.4 | 45.8      | 75.5     | 55.6   | 188| 97.8%|
> >
> > **Table G-4** *LLaVA-1.5-7B — Retain 64 Tokens (88.9%)*
> | Method     | GQA | MMB | MMB-CN  | MME    | POPE | SQA  | VQA (txt) | VQA (v2) | VizWiz | OCRBench | Ratio        |
> |----|---|----|----|------|------|-----|------|---|-----|-----|-----|
> | Vispruner  | 55.1| 58.5| 49.3    | 1362.5 | 83.7 | 67.1 | 40.9| 72.3     | 57.0   | 160| 91.4%|
> | Visionzip  | 54.7| 58.7| 49.7    | 1353.5 | 79.4 | 67.3 | 40.3| 71.3| 56.2   | 155| 90.5%|
> | CDPruner   | 58.1| 59.7| 49.4    | 1373.1 | 84.1 | 67.8 | 41.7| 72.5| 55.1   | 166      | 92.4%|
> | SPLIT | 59.0| 60.1| 49.8    | 1375.5 | 84.5 | 68.6 | 41.5| 73.2| 55.9   | 169| 92.8%|
>
> > **Table G-5** *LLaVA-Next-7B POPE — Retain 320 Tokens (88.9%)*
> | Method | Total Time |
> |-----|------|
> | Vispruner  | 26:46  |
> | Visionzip  | 26:55  |
> | CDPruner   | 30:15  |
> | SPLIT   | 25:52     |
>
> ---
> We are truly grateful for the time you dedicated to our work. We hope that our rebuttal sufficiently addresses your feedback, and we look forward to any further discussion as needed.

---

### Official Review · Reviewer_ZTt1 · 2025-10-30

**Soundness:** 4
**Presentation:** 3
**Contribution:** 3
**Rating:** 6
**Confidence:** 4

**Summary:**

This paper proposed SPLIT, a novel training-free vision tokens selection framework, which first measures token importance via cross-layer hidden-state shifts instead of attention, then allocates adaptive region-level budgets to ensure local coverage, and selects tokens with a simple diversity score to avoid redundancy. On image/video benchmarks, it consistently matches or surpasses SOTA; with LLaVA-1.5-7B it retains >99% accuracy at 192 tokens and ~92.5% at 64, delivering scalable, attention-free token reduction without sacrificing accuracy.

**Strengths:**

1. The proposed SPLIT is training-free and can be seamlessly integrated into several different architectures of vision–language models. The method is novel and clearly reproducible.

2. The experiments are comprehensive and convincing on various benchmarks, models, and tasks, and maintain a great trade-off between accuracy and efficiency.

3. The paper is well-organized, with intuitive figures and sufficient experimental tables, and is easy to follow.

**Weaknesses:**

1. Baseline comparison with other SOTA methods. More methods should be compared, for example, the VisPruner[1], CDPruner[2], and Visionzip[3] in video.

2. Small typo: In the appendix, lines 842, 869, 896, 923, the token numbers are duplicated; please check it again.

[1] Zhang, Q., Cheng, A., Lu, M., Zhuo, Z., Wang, M., Cao, J., ... & Zhang, S. (2024). [CLS] Attention is All You Need for Training-Free Visual Token Pruning: Make VLM Inference Faster. ICCV, 2025.

[2] Zhang, Q., Liu, M., Li, L., Lu, M., Zhang, Y., Pan, J., ... & Zhang, S. (2025). Beyond Attention or Similarity: Maximizing Conditional Diversity for Token Pruning in MLLMs. NeurIPS, 2025.

[3] Senqiao Yang, Yukang Chen, Zhuotao Tian, Chengyao Wang, Jingyao Li, Bei Yu, and Jiaya Jia. Visionzip: Longer is better but not necessary in vision language models. CVPR, 2025.

**Questions:**

1. In the video experiments, you appear to compute token salience and diversity within each frame independently using the same approach as in the image setting. Does SPLIT explicitly account for temporal relationships or cross-frame consistency when selecting tokens?

---

> ### Author Response · Authors · 2025-11-21
> **Response to Reviewer ZTt1**
>
> We sincerely thank the reviewer **ZTt1** for their thoughtful and constructive feedback, and provide our responses below.
>
> ### **A1) Baseline comparison with other SOTA methods**
> >- We conducted __*video understanding*__ experiments with VisPruner, CDPruner, and VisionZip. The results are presented in Table H1-2.
> >- The experiments include: 1) applying the *layer-wise shift*, 2) applying *temporal shift per frame*, and 3) applying *both together*.
> >
> > **Table H-1** *Retain 64×32 Tokens (81.1%)*
> | Method   | MLVU | MVtest | LongB-val| LongB-percept | LongB-relation| MME-w/o sub| MME-short| MME-medium | MME-long| ratio|
> |---|---|---|---|----|---|---|----|---|---|---|
> | Vispruner| 62.1  | 51.2| 56.6  | 62.2 | 51.3 |59.3| 71.5| 55.4| 48.8   | 92.9|
> | Visionzip| 61.9  | 50.8| 56.4  | 61.9 | 51.1 |59.1| 71.4| 55.1| 48.5| 92.5|
> | CDPruner| 62.1  | 52.7| 57.4  | 62.5 | 52.1 |60.2| 72.7| 56.2| 49.1| 94.1|
> | **SPLIT(layer)**   | 62.0  | 52.9| 57.4  | 62.7 | 52.4 |60.5| 72.3| 56.2| 49.4| **94.2**|
> | **SPLIT(frame)**  | 62.4  | 52.9| 57.3  | 62.3 | 51.8 |60.5| 73.0| 56.6| 49.2| **94.2**|
> | **SPLIT(both)**  | 62.3  | 53.0|57.4| 62.6 | 52.4 |60.7| 72.9| 56.6| 49.3| **94.4**|
> >
> > **Table H-2** *Retain 64×16 Tokens (90.5%)*
> | Method| MLVU | MVtest| LongB-val| LongB-percept| LongB-relation| MME-w/o sub| MME-short| MME-medium | MME-long| ratio|
> |---|---|---|--|---|---|---|---|---|---|---|
> | Vispruner| 58.4  | 42.2| 51.2| 56.8 | 45.8 |56.0  | 67.7| 53.1| 47.5| 85.8|
> | Visionzip| 57.9| 41.4| 50.5| 56.3| 45.3 |59.4  | 67.4| 53.3| 47.1| 85.8|
> | CDPruner| 59.4| 49.5| 51.6| 56.9| 46.3 |56.5| 68.3| 53.8| 48.0  | 87.9|
> | **SPLIT(layer)**| 59.1| 49.7| 51.8| 57.1 | 46.1 |56.3  | 68.3| 53.5| 47.6| **87.7**|
> | **SPLIT(frame)**| 59.8| 51.0| 52.1| 57.2 | 47.3 |56.5| 67.2| 53.7| 46.7| **88.1**|
> | **SPLIT(both)**| 59.4| 50.8| 51.6 | 56.9 | 47.0 |56.1| 67.5| 53.7| 46.9| **88.0**|
> ---
> >- We also conducted __*image understanding*__ experiments, with the performance reported in Table H3-5 and the latency in Table H6.
> >
> > **Table H-3** *LLaVA-1.5-7B — Retain 192 Tokens (66.7%)*
> | Method     | GQA  | MMB  | MMB-CN  | MME    | POPE | SQA  | VQA (txt) | VQA (v2) | VizWiz | OCRBench | Ratio|
> |---|---|---|---|---|----|---|-----|---|---|---|---|
> | Vispruner  | 58.6 | 61.5 | 53.6| 1469.9 | 85.2 | 68.3 | 45.7| 75.7| 54.9| 179| 98.0%|
> | Visionzip  | 58.1 | 61.7 | 54.3| 1447.1 | 85.0 | 68.8 | 43.9| 75.7| 54.2| 180| 97.0%|
> | CDPruner   | 59.8 | 62.7 | 55.6| 1483.6 | 86.2 | 67.5 | 48.9| 76.8| 55.7| 189| 99.5%|
> | SPLIT| 59.5 | 62.6 | 54.5| 1479.5 | 85.6 | 68.1 | 46.9| 76.1| 55.3| 192| 99.3%|
> >
> > **Table H-4** *LLaVA-1.5-7B — Retain 128 Tokens (77.8%)*
> | Method     | GQA | MMB | MMB-CN | MME    | POPE  | SQA  | VQA (txt) | VQA (v2) | VizWiz | OCRBench | Ratio|
> |---|---|---|----|----|---|---|----|---|-----|---|----|
> | Vispruner  | 58.0| 61.0| 52.9| 1445.7 | 85.7 | 67.8 | 43.6| 75.0| 55.7 | 177| 96.7%|
> | Visionzip  | 56.9| 61.7| 53.6| 1412.7 | 83.7 | 68.4 | 43.2| 74.5| 55.5 | 184| 95.4%       |
> | CDPruner   | 58.4 | 61.8 | 53.5  | 1458.0 | 85.6 | 66.8 | 45.2 | 75.1 | 55.2 | 180 | 97.4% |
> | SPLIT | 58.8| 62.5| 51.9    | 1455.2 | 85.5 | 68.4 | 45.8| 75.5| 55.6   | 188| 97.8%|
> >
> > **Table H-5** *LLaVA-1.5-7B — Retain 64 Tokens (88.9%)*
> | Method     | GQA | MMB | MMB-CN  | MME    | POPE | SQA  | VQA (txt) | VQA (v2) | VizWiz | OCRBench | Ratio  |
> |---|---|---|---|---|---|---|---|---|---|---|---|
> | Vispruner  | 55.1| 58.5| 49.3  | 1362.5 | 83.7 | 67.1 | 40.9  | 72.3    | 57.0   | 160| 91.4%    |
> | Visionzip  | 54.7| 58.7| 49.7  | 1353.5 | 79.4 | 67.3 | 40.3  | 71.3   | 56.2   | 155| 90.5%  |
> | CDPruner   | 58.1| 59.7| 49.4  | 1373.1 | 84.1 | 67.8 | 41.7  | 72.5   | 55.1   | 166      | 92.4% |
> | SPLIT| 59.0| 60.1| 49.8  | 1375.5 | 84.5 | 68.6 | 41.5 | 73. | 55.9  | 169      | 92.8%      |
>
> > **Table H-6** *LLaVA-Next-7B POPE — Retain 320 Tokens (88.9%)*
> | Method | Total Time |
> |---|---|
> | Vispruner| 26:46|
> | Visionzip| 26:55 |
> | CDPruner| 30:15|
> | SPLIT | 25:52 |
>
> ---
> ### **A2) Small typo: In the appendix**
> >- Thank you for the careful observation. We have fixed the typos in the revised manuscript.
>
> ---
> ### **Q1) cross-frame consistency in video token selection**
> >- As noted in A1, we included experiments that apply temporal shifts across frames.
>
> ---
> We sincerely thank you once again for your insightful comments and dedicated effort in reviewing our paper. We are hopeful that this rebuttal resolves all the concerns raised.

---

### Official Review · Reviewer_xWpK · 2025-11-01

**Soundness:** 3
**Presentation:** 3
**Contribution:** 2
**Rating:** 2
**Confidence:** 3

**Summary:**

The paper introduces SPLIT-VLM, a framework that makes VLMs more efficient by intelligently dropping redundant visual tokens. It solves the limitations of prior methods by uniquely balancing token importance with spatial coverage. The method partitions an image, assigning budgets to regions using a novel temporal shift importance metric to guarantee full coverage . Within each budget, a robust diversity score selects the most informative tokens.

**Strengths:**

1.	This paper proposes a novel, attention-free metric to efficiently measure token importance.
2.	This paper introduces a strategy that guarantees full image coverage by assigning token budgets to different regions.
3.	This paper proposes a robust diversity score-based method for selecting the most representative tokens without sensitive hyperparameters.

**Weaknesses:**

1.	The performance improvement of the proposed method is rather marginal. Compared with DivPrune and DART, it shows no significant advantage in either performance (as is particularly evident in Table 6 and 7) or efficiency.
2.	The ablation study is not sufficiently convincing. For example, in Figure 5, it is difficult to discern any clear advantage of the Adaptive strategy over HiRED.
3.	There is a lack of clear results or examples demonstrating that the proposed local budget allocation combined with the diversity score effectively compensates for the information that previous methods tend to miss, thereby improving understanding performance.

**Questions:**

1.	In lines 415-417, the authors claim that random selection achieves performance comparable to HiRED. This statement is confusing, as it is not clear how such a conclusion can be drawn from Figure 5.
2.	The importance of global coverage for understanding tasks remains unclear-what if the informative content of an image is concentrated in only a specific region? The paper lacks concrete examples or results to substantiate the claimed superiority of the proposed method over previous approaches in this respect.
3.	Why is the parameter lambda in Equation 10 set to 0.5? How would changing its value affect the results?

---

> ### Author Response · Authors · 2025-11-20
> **Response to Reviewer xWpK (1)**
>
> We thank Reviewer **xWpK** for the thoughtful feedback, and we provide our detailed responses below.
>
> ### **A1) Limited improvement and no clear advantage over DivPrune and DART**
> >- Although SPLIT shows slightly lower performance only in a very small number of benchmarks and only under specific budget conditions, it maintains consistently stable performance across virtually all other settings and achieves substantially higher performance under small token budgets. In addition, SPLIT achieves the fastest latency in all evaluated conditions, and this advantage becomes more pronounced as the token budget increases. SPLIT shows a consistent strength: **better performance under smaller budgets and better latency under larger budgets**.
> > - Table C presents the latency across different token budgets.
>
> > **Table C** *LLaVA-Next-7B latency (88.9%)*
> | Method |Tokens| Total Time|Δ Time vs 320 |
> |---------|---------|-----------|---------|
> | GreedyPrune | 320 | 31:17 | - |
> | GreedyPrune | 640 | 32:08 | +0:51 |
> | GreedyPrune | 960 | 33:06 | +1:49 |
> | DART | 320 | 27:28 | - |
> | DART | 640 | 28:13 | +0:45 |
> | DART | 960 | 28:57 | +1:29 |
> | SPLIT | 320 | 25:52 | - |
> | SPLIT | 640 | 26:24 | +0:32 |
> | SPLIT | 960 | 26:58 | +1:06 |
>
>
> ---
> ### **A2) Adaptive budget allocation shows little benefit over HiRED**
> > - We thank the reviewer for the helpful comment. We believe the concern arises from a misunderstanding of the intention behind Figure 5, and we appreciate the opportunity to clarify it.
> >>- HiRED is included simply as a reference method, and it inherently applies both importance estimation and its own token-selection strategy.
> >>- We apply random sampling to isolate budget allocation, so comparing adaptive allocation with HiRED is not fair since HiRED keeps its own selection mechanism. The reviewer’s concern therefore arises from a comparison that is not appropriate in this setting.
> >>- When we replace random selection with our full diversity-score based selection (_as used in the main experiments_), the performance gap over HiRED becomes clear.
>
> ---
> ### **A3) Lack of clear results or examples demonstrating that the proposed local budget allocation**
> >- To more clearly showcase the benefits of our budget allocation strategy, we additionally perform experiments. **We enforced a local budget across multiple methods**(as marked with * in the table), and the results are shown in Table D.
> >- **We further quantified the unevenness of region-wise token distributions** for each pruning method under both the non-allocation and allocation settings by computing the **Max/Mean ratio** $M(B)= \frac{\max_k B_k}{\frac{1}{K}\sum_{k=1}^{K} B_k }.$ and the **Gini coefficient** $G(B)= \frac{\sum_{i=1}^{K} \sum_{j=1}^{K} \left| B_i - B_j \right|}{2K\sum_{k=1}^{K} B_k}$.
>
> >**Table D** *LLaVA-1.5-7B — Retain 64 Tokens (88.9%)*
> | Method     | GQA | MMB | MMB-CN  | MME    | POPE | SQA  | VQA (txt) | VQA (v2) | VizWiz | OCRBench | Ratio  |$M(B)$|$G(B)$|
> |------------|-----|-----|---------|--------|------|------|-----------|----------|--------|-------|-------|---|---|
> | Vispruner  | 55.1| 58.5| 49.3    | 1362.5 | 83.7 | 67.1 | 40.9      | 72.3     | 57.0   | 160      | 91.4%     | 2.75| 0.384|
> | Vispruner* | 55.5| 58.9| 49.7    | 1365.5 | 83.9 | 67.3 | 41.1      | 72.8     | 57.3   | 162      | 91.7%     | 2.50| 0.358|
> | Visionzip  | 54.7| 58.7| 49.7    | 1353.5 | 79.4 | 67.3 | 40.3      | 71.3     | 56.2   | 155      | 90.5%        |2.60| 0.377|
> | Visionzip* | 55.0| 59.4| 50.1    | 1360.4 | 80.9 | 67.6 | 40.6      | 71.9     | 56.7   | 160      | 91.0%        | 2.50| 0.346|
> | CDPruner   | 58.1| 59.7| 49.4    | 1373.1 | 84.1 | 67.8 | 41.7      | 72.5     | 55.1   | 166      | 92.4%     | 2.45| 0.325|
> | CDPruner*  | 58.9| 60.3| 50.7    | 1375.6 | 84.9 | 68.1 | 42.1      | 73.1     | 56.3   | 170      | 93.0%     | 2.20|  0.269|

---

> ### Author Response · Authors · 2025-11-20
> **Response to Reviewer xWpK (2)**
>
> We thank Reviewer **xWpK** for the thoughtful feedback, and we provide our detailed responses below.
>
> ### **Q1) Unclear how random is comparable with HiRED**
> >- Thank you for pointing out that the wording may have been confusing. *The revised manuscript (PDF) has been uploaded, in which the change(lines 415-417) is highlighted in red for your convenience.*
>
> ---
> ### **Q2) The importance of global coverage for understanding tasks remains unclear**
> >- As explained in the manuscript, although some images may concentrate most of their salient content in a specific region, many vision–language understanding tasks require relational or contextual cues distributed across multiple regions, making diversity-aware region selection essential [1][2].
> >- To address the reviewer’s concern we provide a direct visual example. As shown in [link1], **the temporal shift extracted from early layers captures highly salience patches**.
> >- However, as reported in Figure 6 in manuscript, using only early-layer or middle-layer salience for pruning leads to performance degradation. Considering the overall change rate instead results in better performance, which **highlights the strength of approaches that preserve global coverage**.
>
> ---
> ### **Q3) The importance of global coverage for understanding tasks remains unclear**
> >- One of our key motivations (_Lim_. 4 in manuscript) was to develop a method that avoids the hyperparameter sensitivity issues observed in prior diversity-based approaches. For this reason, we selected λ = 0.5 as a fixed and intuitive default by the following considerations:
> >>- Setting λ = 0 reduces Eq. (10) to pure redundancy suppression, ignoring token distinctiveness.
> >>- λ =1 excessively amplifies σᵢ, leading to selection of outlier-like tokens that carry limited semantic value.
> >-  Following the reviewer’s valuable feedback, we conducted an additional sensitivity study with λ ∈ {0.3, 0.4, 0.5, 0.6, 0.7}. Results are summarized in Table E.
>
> > **Table E** *LLaVA-1.5-7B — Retain 64 Tokens (88.9%)*
> | Method      | GQA | MMB | MMB-CN  | MME    | POPE | SQA  | VQA (txt) | VQA (v2) | VizWiz | OCRBench | Ratio        |
> |------------ |-----|-----|---------|--------|------|------|-----------|----------|--------|----------|--------------|
> | SPLIT(0.3)  | 58.1| 59.4| 50.1    | 1374.1 | 84.1 | 67.9 | 42.1      | 72.1     | 56.2   | 168      | 92.6%        |
> | SPLIT(0.4)  | 58.4| 59.7| 49.9    | 1376.4 | 84.9 | 68.1 | 42.6      | 74.1     | 55.5   | 167      | 92.8%        |
> | SPLIT(0.5)  | 59.0| 60.1| 49.8    | 1375.5 | 84.5 | 68.6 | 41.5      | 73.2     | 55.9   | 169      | 92.8%        |
> | SPLIT(0.6)  | 59.1| 60.3| 49.1    | 1371.6 | 85.1 | 68.1 | 41.7      | 73.9     | 55.1   | 166      | 92.5%        |
> | SPLIT(0.7)  | 57.1| 59.4| 50.1    | 1368.4 | 83.7 | 67.5 | 40.1      | 72.4     | 55.1   | 164      | 91.9%        |
>
> ---
> ### References ###
> > [1] Wen, Z., Gao, Y., Wang, S., Zhang, J., Zhang, Q., Li, W., Zhang, L. Stop looking for important tokens in multimodal language models: Duplication matters more. arXiv 2025.
> >
> > [2] Zhang Q, Liu M, Li L, et al. Beyond Attention or Similarity: Maximizing Conditional Diversity for Token Pruning in MLLMs. NeurIPS 2025
> >
> > [Link1] https://github.com/vlm15400/iclr2026-15400/blob/main/fig_temporal_shift.png
>
> Thank you again for your valuable feedback. We hope our rebuttal address your concerns. If you have further questions, please do not hesitate to let us know.

---

### Official Review · Reviewer_uYns · 2025-11-02

**Soundness:** 3
**Presentation:** 2
**Contribution:** 2
**Rating:** 2
**Confidence:** 5

**Summary:**

This paper targets VLM inference efficiency. It first analyzes the limitations of prior saliency-based and diversity-based approaches, then proposes SPLIT, a framework that combines both ideas. The paper provides a theoretical analysis of the method and conducts extensive experiments across different model architectures and multimodal tasks. Results show that SPLIT outperforms selected baselines, with additional efficiency and ablation studies.

**Strengths:**

1.	The paper presents both theoretical and empirical perspectives. The proposed method has a theoretical basis and shows solid performance.
2.	The experiments are extensive and cover multiple VLM architectures and multiple multimodal tasks.
3.	The paper provides detailed efficiency analysis, showing that the method brings practical improvements to VLM inference efficiency.

**Weaknesses:**

1.	The paper lacks discussion of key related work. To avoid LLM attention bias, methods such as VisionZip[1] and VisPruner[2] prune after the vision encoder. Other methods such as CDPruner[3] and MoB[4] also combine saliency and diversity. These works are not discussed or compared.
2.	The paper does not demonstrate how uneven token-budget distribution across regions affects pruning performance, which weakens the motivation.
3.	The metrics used for adaptive region-level budget allocation are highly heuristic. The paper does not explain why tokens with high change rate should be considered more important, and no visualization analysis is provided.

[1] Yang S, Chen Y, Tian Z, et al. Visionzip: Longer is better but not necessary in vision language models. CVPR 2025.
[2] Zhang Q, Cheng A, Lu M, et al. Beyond text-visual attention: Exploiting visual cues for effective token pruning in vlms. ICCV 2025.
[3] Zhang Q, Liu M, Li L, et al. Beyond Attention or Similarity: Maximizing Conditional Diversity for Token Pruning in MLLMs. NeurIPS 2025.
[4] Li Y, Zhan H, Chen T, et al. Why 1+ 1< 1 in Visual Token Pruning: Beyond Naive Integration via Multi-Objective Balanced Covering. arXiv 2025.
[5] Li A, Duan Y, Zhang J, et al. TransPrune: Token Transition Pruning for Efficient Large Vision-Language Model. arXiv 2025.

**Questions:**

1.	The idea of allocating budget based on change rate is very similar to TransPrune[5]. Can the authors clarify the difference?
2.	In budget allocation, how is the sum of token-budget allocations guaranteed to equal B? In Eq. (7), the sum appears to be 2B. Is a factor of 1/2 missing?

---

> ### Author Response · Authors · 2025-11-20
> **Response to Reviewer uYns (1)**
>
> We sincerely thank reviewer **uYns** for the valuable comments and constructive feedback. Please find our point-by-point response below.
>
> ### **A1) Lack of discussion on key related tasks**
> >
> >- Table A1-A4 reports image understanding results for VisionZip, VisPruner, and CDPruner. We carefully examined the official GitHub repositories of all three methods and ensured their algorithms were faithfully reproduced. All comparisons were conducted under the same experimental environment, and we evaluated every method using the LMMs-eval. Table A5 additionally reports the latency measurements.
>
> >- And following the reviewer’s comment, we clarify the differences from related work. We will update it after the discussion concludes.
> >>- VisPruner selects tokens with high attention scores and then removes redundancy, while VisionZip uses attention and similarity to pick informative tokens. However, both methods still fundamentally rely on attention scores (*Lim*.1).
> >>- CDPruner captures diversity well but relies on DPP, which introduces heavy computational overhead. Also, it requires an additional CLIP text encoder for instruction-conditioned similarity, **making it impractical for models like Qwen**.
> >>- MoB balances visual preservation(VP) and prompt alignment(PA) using a Hausdorff-distance formulation to improve global coverage.
> >>- These methods provide strong ideas for global coverage and diversity, but **they inevitably add substantial computation**. In contrast, SPLIT achieves both coverage and diversity through lightweight region-wise budget allocation and an efficient diversity score.
>
> > **Table A-1** *LLaVA-1.5-7B — Full Token, 576 Tokens (100%)*
> | Method  | GQA   | MMB   | MMB-CN | MME     | POPE | SQA   | VQA (txt) | VQA (v2) | VizWiz | OCRBench | Ratio  |
> |---------|-------|-------|--------|---------|------|-------|-----------|----------|--------|----------|--------|
> | Vanilla | 61.2  | 63.1  | 54.8   | 1477.7  | 85.4 | 68.1  | 48.7      | 77.4     | 53.8   | 205      | 100%   |
> >
> > **Table A-2** *LLaVA-1.5-7B — Retain 192 Tokens (66.7%)*
> | Method     | GQA  | MMB  | MMB-CN  | MME    | POPE | SQA  | VQA (txt) | VQA (v2) | VizWiz | OCRBench | Ratio        |
> |------------|------|------|---------|--------|------|------|-----------|----------|--------|----------|--------------|
> | Vispruner  | 58.6 | 61.5 | 53.6    | 1469.9 | 85.2 | 68.3 | 45.7      | 75.7     | 54.9   | 179      | 98.0%        |
> | Visionzip  | 58.1 | 61.7 | 54.3    | 1447.1 | 85.0 | 68.8 | 43.9      | 75.7     | 54.2   | 180      | 97.0%        |
> | CDPruner   | 59.8 | 62.7 | 55.6    | 1483.6 | 86.2 | 67.5 | 48.9      | 76.8     | 55.7   | 189      |99.5% |
> | SPLIT| 59.5 | 62.6 | 54.5    | 1479.5 | 85.6 | 68.1 | 46.9      | 76.1     | 55.3   | 192      | 99.3%        |
> >
> > **Table A-3** *LLaVA-1.5-7B — Retain 128 Tokens (77.8%)*
> | Method     | GQA | MMB | MMB-CN | MME    | POPE  | SQA  | VQA (txt) | VQA (v2) | VizWiz | OCRBench | Ratio        |
> |------------|-----|-----|---------|--------|------|------|-----------|----------|--------|----------|--------------|
> | Vispruner  | 58.0| 61.0| 52.9    | 1445.7 | 85.7 | 67.8 | 43.6      | 75.0     | 55.7   | 177      | 96.7%        |
> | Visionzip  | 56.9| 61.7| 53.6    | 1412.7 | 83.7 | 68.4 | 43.2      | 74.5     | 55.5   | 184      | 95.4%        |
> | CDPruner   | 58.4 | 61.8 | 53.5  | 1458.0 | 85.6 | 66.8 | 45.2 | 75.1 | 55.2 | 180 | 97.4% |
> | SPLIT| 58.8| 62.5| 51.9    | 1455.2 | 85.5 | 68.4 | 45.8      | 75.5     | 55.6   | 188      | 97.8%      |
> >
> > **Table A-4** *LLaVA-1.5-7B — Retain 64 Tokens (88.9%)*
> | Method     | GQA | MMB | MMB-CN  | MME    | POPE | SQA  | VQA (txt) | VQA (v2) | VizWiz | OCRBench | Ratio        |
> |------------|-----|-----|---------|--------|------|------|-----------|----------|--------|----------|--------------|
> | Vispruner  | 55.1| 58.5| 49.3    | 1362.5 | 83.7 | 67.1 | 40.9      | 72.3     | 57.0   | 160      | 91.4%        |
> | Visionzip  | 54.7| 58.7| 49.7    | 1353.5 | 79.4 | 67.3 | 40.3      | 71.3     | 56.2   | 155      | 90.5%        |
> | CDPruner   | 58.1| 59.7| 49.4    | 1373.1 | 84.1 | 67.8 | 41.7      | 72.5     | 55.1   | 166      | 92.4%        |
> | SPLIT| 59.0| 60.1| 49.8    | 1375.5 | 84.5 | 68.6 | 41.5      | 73.2     | 55.9   | 169      | 92.8%  |
>
> > **Table A-5** *LLaVA-Next-7B POPE — Retain 320 Tokens (88.9%)*
> | Method     | Total Time |
> |------------|------------|
> | Vispruner  | 26:46      |
> | Visionzip  | 26:55      |
> | CDPruner   | 30:15      |
> | SPLIT      | 25:52      |

---

> ### Author Response · Authors · 2025-11-20
> **Response to Reviewer uYns (2)**
>
> We sincerely thank reviewer **uYns** for the valuable comments and constructive feedback. Please find our point-by-point response below.
>
> ### **A2) Demonstrate how uneven token-budget distribution across regions affects pruning performance**
> >- **We enforced a region-wise budget across multiple methods**(as marked with * in the table), and the results are shown in Table B. All methods showed consistent performance improvements. This confirms that even a minimal local coverage can stabilize or boost performance.
> >- **We further quantified the unevenness of region-wise token distributions** for each pruning method under both the non-allocation and allocation settings by computing the **Max/Mean ratio** $M(B)= \frac{\max_k B_k}{\frac{1}{K}\sum_{k=1}^{K} B_k },$ and the **Gini coefficient** $G(B)= \frac{\sum_{i=1}^{K} \sum_{j=1}^{K} \left| B_i - B_j \right|}{2K\sum_{k=1}^{K} B_k}$.
>
> >**Table B** *LLaVA-1.5-7B — Retain 64 Tokens (88.9%)*
> | Method     | GQA | MMB | MMB-CN  | MME    | POPE | SQA  | VQA (txt) | VQA (v2) | VizWiz | OCRBench | Ratio  |$M(B)$|$G(B)$|
> |------------|-----|-----|---------|--------|------|------|-----------|----------|--------|-------|-------|---|---|
> | Vispruner  | 55.1| 58.5| 49.3    | 1362.5 | 83.7 | 67.1 | 40.9      | 72.3     | 57.0   | 160      | 91.4%     | 2.75| 0.384|
> | Vispruner* | 55.5| 58.9| 49.7    | 1365.5 | 83.9 | 67.3 | 41.1      | 72.8     | 57.3   | 162      | 91.7%     | 2.50| 0.358|
> | Visionzip  | 54.7| 58.7| 49.7    | 1353.5 | 79.4 | 67.3 | 40.3      | 71.3     | 56.2   | 155      | 90.5%        |2.60| 0.377|
> | Visionzip* | 55.0| 59.4| 50.1    | 1360.4 | 80.9 | 67.6 | 40.6      | 71.9     | 56.7   | 160      | 91.0%        | 2.50| 0.346|
> | CDPruner   | 58.1| 59.7| 49.4    | 1373.1 | 84.1 | 67.8 | 41.7      | 72.5     | 55.1   | 166      | 92.4%     | 2.45| 0.325|
> | CDPruner*  | 58.9| 60.3| 50.7    | 1375.6 | 84.9 | 68.1 | 42.1      | 73.1     | 56.3   | 170      | 93.0%     | 2.20|  0.269|
>
> ---
> ### **A3) Explain why tokens with high change rate should be considered more important**
> >- Prior work [1] shows that attention scores do not reliably represent importance and are incompatible with efficient kernels (e.g., FlashAttention). In contrast, large temporal shift intuitively indicates that a token is receiving stronger attention.
> >- The visualization [link1] compares HiRED’s attention-score maps with our temporal shift. **Temporal shift highlights salient regions in early–mid layers and broader scene context in later layers**. Attention scores closely follow mid-layer patterns.
> >- As shown in Fig. 6, relying solely on mid-layer signals that correspond to attention scores yields lower performance than using changes from all layers, **indirectly indicating that temporal shift reflects token importance more faithfully than attention**. If needed, we can also report results where region budgets are computed directly from attention scores.
>
> ---
> ### **Q1) Clarify the difference with TransPrune**
> >- Although both works refer to “changes” in hidden states, the methods differ fundamentally in purpose, location, and metric design.
> >>- __*Magnitude metric*__: TransPrune uses a scale-ratio magnitude metric. This measures how much the representation grows or shrinks within LLM attention/FFN layers. SPLIT uses a difference-based magnitude metric. This measures the absolute update size across ViT layers, which correlates with patch informativeness.
> >>- __*Purpose*__: TransPrune uses transition signals as the final token-importance score for pruning. SPLIT uses temporal shift only as a coarse regional budget signal; actual token selection is done by diversity, so additional operations like direction shift are excessive.
> >>- __*Location*__: TransPrune measures transitions in LLM attention/FFN. SPLIT measures ViT layer-to-layer patch updates.
>
> ---
> ### **Q2) Issues in the budget allocation formula**
> >- Thank you for pointing this out. Eq. (7) provides raw region-level scores, and in our implementation these values are rescaled so that their final sum becomes exactly B. We acknowledge the confusion and will clarify this more explicitly in the revised version.
>
> ---
>
> ### **References** ###
> > [1] Guo, Z., Kamigaito, H., & Watanabe, T.  Attention score is not all you need for token importance indicator in kv cache reduction: Value also matters. arXiv 2024.
> >
> > [Link1] https://github.com/vlm15400/iclr2026-15400/blob/main/fig_temporal_shift.png
>
> Thank you once again for your effort in reviewing our paper. We hope our rebuttal will address your concerns. We are glad to have further discussions with you.

---

### Author Response · Authors · 2025-11-28

**Dear AC and Reviewers**,

Thank you again for your time and for coordinating the discussion phase. We have carefully addressed all reviewer comments in detail in our recent rebuttal, and we hope that our clarifications are helpful for resolving the confusion.

If possible, we would sincerely appreciate it if you could take another look at our responses when you have a moment. We would be happy to provide further clarification on any remaining concerns.

Thank you very much for your consideration and for your efforts during the discussion phase.

---

### Author Response · Authors · 2025-12-04
**Final Clarification for the Area Chair (1/4)**

Dear Area Chair,

We sincerely appreciate your efforts in guiding the review process despite the unexpected reassignment. We hope that the additional clarifications below will assist your evaluation of our submission.

---
## **Brief Summary**
Across all reviewers, there were no concerns regarding the novelty, correctness, or motivation of our work. The main points raised focused on:

1) comparisons to additional SOTA methods  (Reviewers **uYns**, **ZTt1**, **96wB**),

2)  justification for the temporal-shift importance metric (Reviewers **uYns**),

3) clarification of our adaptive region-level budgeting (Reviewers **uYns**, **xWpK**),

4) interpretation of specific ablations (Reviewers **uYns**, **xWpK**), and minor formula details (Reviewers **ZTt1**, **96wB**).

---

> ### Author Response · Authors · 2025-12-04
> **Final Clarification for the Area Chair (2/4)**
>
> ## **Reviewer uYns (No Response)**
> #### **W1: Lack of discussion on key related tasks**
>
>  - We clarified SPLIT’s distinct mechanism—temporal shift-based importance, region-level budget allocation, and distribution-aware diversity scoring—and explicitly articulated how each component differs from existing saliency- or diversity-based pruning methods and addresses their limitations. We further provided direct experimental comparisons against VisionZip[1], VisPruner[2], and CDPruner[3] in **Tables A-1** through **A-4**, along with latency evaluations reported in **Tables A–5**, **demonstrating SPLIT’s consistent advantage across accuracy and efficiency**. We also broadened our discussion to include more recent approaches, such as MoB[4] and TransPrune[5], further clarifying SPLIT’s distinctive contributions and their relation to these methods.
>
> #### **W2: Demonstrate how uneven token-budget distribution across regions affects pruning performance**
>
>  - We enforced our region-level budget allocation strategy across multiple baselines (**Table B**). We **observed consistent performance gains in all methods**, proving that "_local coverage_" is a generalizable contribution beyond our specific model. We also quantified token imbalance (**_e.g._,Gini coefficient**), demonstrating that adaptive budget allocation plays a decisive role in stabilizing performance.
>
> #### **W3: Explain why tokens with a high change rate should be considered more important**
>
>  - We showed that attention scores are unreliable and incompatible with efficient kernels, while temporal shift reflects meaningful representational updates across layers. Visualizations reveal that the shift highlights salient regions in early–mid layers and scene-level cues in deeper layers. Pruning based solely on mid-layer signals—where attention dominates—yields inferior performance, demonstrating that **temporal shift provides a more robust importance signal**.
>
> #### **Q1: Clarify the difference with TransPrune**
>
>  - We explicitly showed that TransPrune[5] measures scale-ratio transitions inside LLM attention/FFN layers and uses them directly for pruning, whereas SPLIT measures difference-based semantic displacement across ViT layers and uses it only for regional budgeting.
>
>
> Regarding the concerns raised by Reviewer **uYns**, we provided detailed explanations, refined discussions, and complementary experiments. We are confident that these revisions effectively resolve all issues they identified.
>
> ---
> ## **Reviewer xWpK (No Response)**
> #### **W1: Limited improvement and no clear advantage over DivPrune and DART**
>  - We highlighted that SPLIT’s accuracy is slightly lower only in a few cases, while remaining stable across nearly all benchmarks. SPLIT shows clear strength under extremely small budgets—where pruning matters most—and, as demonstrated in the latency results (**Table C**), **provides the best accuracy–latency tradeoff across all settings**.
>
> #### **W2: Adaptive budget allocation shows little benefit over HiRED**
>  - The concern originates from a misunderstanding of Figure 5. In this ablation, we isolated the effect of budgeting using random sampling; HiRED applies its own selection mechanism and is not a comparable control. When applying SPLIT’s full selection pipeline, its advantage over HiRED becomes clear.
>
> #### **W3: Lack of clear results or examples demonstrating that the proposed local budget allocation**
>  - We strengthened our claim by enforcing local coverage across multiple pruning methods (**Table D**). All methods improved when minimal regional coverage was guaranteed, and imbalance metrics (**Max/Mean**, **Gini**) confirmed that adaptive budgeting prevents extreme skew. These results demonstrate **the concrete benefits of SPLIT’s coverage-aware design**.
>
> #### **Q2: Why global coverage matters even when salience is localized**
>  - Prior work[6][7]  has already shown that vision–language understanding often relies on relational or contextual cues distributed across multiple regions, making diversity-aware region selection essential. To address the reviewer’s concern, we also provided additional visual analyses illustrating [link1] how our method captures these distributed cues more effectively.  In addition, the necessity of global coverage is already clearly articulated in the main manuscript (lines 190–207).
>
> #### **Q3: Why is the parameter lambda in Equation 10 set to 0.5?**
>  - Our ablation study shows that SPLIT remains stable across the λ range of 0.3–0.7 (**Table E**), with λ = 0.5 providing the most balanced trade-off. Values outside this range are undesirable: λ = 0 collapses the score into pure redundancy suppression, while λ = 1 amplifies variance excessively.
>
> Altogether, the extended experiments and visual analyses meaningfully address the concerns raised by Reviewer **xWpK** and clarify the value of our proposed approach.

---

> ### Author Response · Authors · 2025-12-04
> **Final Clarification for the Area Chair (3/4)**
>
> ## **Reviewer ZTt1 (No Response)**
> #### **W1 & Q1: Comparison with SOTA Methods and Cross-Frame Evaluation in Video**
>
>  - We conducted comparison experiments against VisionZip[1], VisPruner[2], and CDPruner[3] and added three complementary video evaluations: (1) **layer-wise shift**, (2) **per-frame shift**, and (3) **combined**. SPLIT consistently matches or surpasses all baselines across multiple video benchmarks (**Tables H1–H2**), even under aggressive budgets (e.g., 64×16), demonstrating stable extension to video without architectural changes.
>
> We also note that Reviewer **ZTt1**’s constructive feedback motivated us to broaden our analysis to cross-frame video evaluation, which offered a valuable new perspective on SPLIT’s temporal behavior. Incorporating this suggestion provided additional insight, and the resulting experiments showed consistently strong performance, further improving the completeness and quality of the upcoming revised version.
>
> ---
> ## **Reviewer 96wB (No Response)**
> #### **W1: Limited performance improvement**
>
>  - While all methods naturally improve as the token budget increases, SPLIT maintains stable accuracy even under the smallest budgets and, as shown in **Table F**, achieves the best accuracy–latency tradeoff across all settings.
>
> #### **W2: Overlap between SPLIT’s diversity selection and prior methods**
>
>  - We explained that SPLIT’s diversity score differs fundamentally: DivPrune uses max–min distance, while SPLIT uses mean–variance structure for stability; CDPruner[3] relies on heavy DPP sampling and requires CLIP-based pretraining alignment. We reproduced all baselines under identical conditions (**Tables G1–G5**), confirming SPLIT’s distinct formulation and competitive performance without added computation.
>
> We are encouraged by Reviewer **96wB**’s positive assessment. The additional empirical evidence and clearer methodological distinctions directly address the points they raised and further substantiate the strengths they noted.

---

> ### Author Response · Authors · 2025-12-04
> **Final Clarification for the Area Chair (4/4)**
>
> ## **Conclusion**
>
> We believe the extensive additional experiments and detailed explanations fully address the reviewers’ concerns and further highlight the robustness and practical effectiveness of SPLIT. In the absence of further discussion from the reviewers, we aimed to preemptively address all raised issues through substantial analyses and additional updates. We hope these additions help convey the strengthened state of the paper in your final assessment.
>
> Best regards,
>
> Authors of Paper 15400
>
> ---
> #### **References**
> [1] Yang S, Chen Y, Tian Z, et al. Visionzip: Longer is better but not necessary in vision language models. CVPR 2025.
>
> [2] Zhang Q, Cheng A, Lu M, et al. Beyond text-visual attention: Exploiting visual cues for effective token pruning in vlms. ICCV 2025.
>
> [3] Zhang Q, Liu M, Li L, et al. Beyond Attention or Similarity: Maximizing Conditional Diversity for Token Pruning in MLLMs. NeurIPS 2025.
>
> [4] Li Y, Zhan H, Chen T, et al. Why 1+ 1< 1 in Visual Token Pruning: Beyond Naive Integration via Multi-Objective Balanced Covering. arXiv 2025.
>
> [5] Li A, Duan Y, Zhang J, et al. TransPrune: Token Transition Pruning for Efficient Large Vision-Language Model. arXiv 2025.
>
> [6] Wen, Z., Gao, Y., Wang, S., Zhang, J., Zhang, Q., Li, W., Zhang, L. Stop looking for important tokens in multimodal language models: Duplication matters more. arXiv 2025.
>
> [7] Zhang Q, Liu M, Li L, et al. Beyond Attention or Similarity: Maximizing Conditional Diversity for Token Pruning in MLLMs. NeurIPS 2025
>
> [Link1] https://github.com/vlm15400/iclr2026-15400/blob/main/fig_temporal_shift.png

---

### Meta-Review · Area_Chair_uZmD · 2026-01-07

**Summary:**

This paper proposes a training-free token pruning framework with an interesting combination of temporal-shift-based importance, region-level budget allocation, and diversity-based selection. While the rebuttal substantially clarified several technical points—especially missing baselines, ablation setups, and formula details—the final evaluation remains mixed.

Two reviewers initially raised serious concerns about the strength and clarity of the empirical evidence. Although many of these issues were addressed in the rebuttal, the improvements primarily strengthened the paper’s positioning rather than clearly elevating its empirical impact. In particular, performance gains over strong baselines remain modest in most medium and large budget regimes, and the core temporal-shift importance metric, while better motivated, still appears somewhat heuristic.

The other two reviewers were already moderately positive, but did not indicate that the additional results would meaningfully change their assessment. Overall, despite a careful and thorough rebuttal, the paper did not reach a level of consensus support among reviewers that would justify acceptance at ICLR, leading to the recommendation to reject.

**Reviewer Concerns:**

Reviewer Concerns

Addressed concerns:
For uYns and xWpK, the major technical concerns raised in the initial reviews were largely addressed. The authors added missing SOTA comparisons (VisionZip, VisPruner, CDPruner), clarified the role of temporal-shift importance and its distinction from TransPrune, fixed the budget-allocation formula ambiguity, and provided clearer ablations showing the effect of region-level budgeting across multiple baselines. Additional latency results, λ sensitivity analysis, and video experiments further reduced the original sources of confusion.

Outstanding concerns:
For uYns and xWpK, some skepticism may remain regarding the magnitude of gains at medium or large token budgets, and the temporal-shift metric, while better motivated and visualized, still has a heuristic aspect. These issues were mitigated but not fully eliminated.

For ZTt1 and 96wB, no substantial concerns remain. Their requested additions and minor issues (extra baselines, video evaluation, typos, overlap clarification) were addressed, but there was no indication that these changes would alter their original, already positive, assessments.

**Reviewer Scores:**

Reviewer uYns

Original score: 2

Expected score after discussion: 4

Rationale: The main concerns (missing SOTA comparisons, motivation of temporal shift, budget imbalance, TransPrune confusion, and Eq.7) were directly addressed with new experiments, clarifications, and fixes. Although the reviewer did not explicitly respond or promise a score increase, the rebuttal substantially reduced the original major concerns. A moderate +2 adjustment is reasonable given the detailed and concrete feedback provided by the authors.

Reviewer xWpK

Original score: 2

Expected score after discussion: 4

Rationale: Key issues around ablation clarity, local budget effectiveness, global coverage motivation, and λ sensitivity were all explicitly clarified with additional experiments and explanations. While performance gains at larger budgets remain modest, most core misunderstandings were resolved. In the absence of explicit reviewer confirmation, a conservative +2 increase is appropriate.

Reviewer ZTt1

Original score: 6

Expected score after discussion: 6

Rationale: The reviewer was already positive. Requested additions (extra baselines, video experiments, typo fixes) were completed, but there was no indication that the score would increase further.

Reviewer 96wB

Original score: 6

Expected score after discussion: 6

Rationale: Concerns were mostly minor and addressed (formatting, overlap clarification, efficiency analysis). The reviewer already rated the paper above the acceptance threshold and did not signal a higher score.

Score Summary

Final scores: 4, 4, 6, 6

Average score: 5.0

---

### Decision · Program_Chairs · 2026-01-26

Reject